# Virological, Serological and Clinical Analysis of Chikungunya Virus Infection in Thai Patients

**DOI:** 10.3390/v14081805

**Published:** 2022-08-18

**Authors:** Yin May Tun, Prakaykaew Charunwatthana, Chatnapa Duangdee, Jantawan Satayarak, Sarocha Suthisawat, Oranit Likhit, Divya Lakhotia, Nathamon Kosoltanapiwat, Passanesh Sukphopetch, Kobporn Boonnak

**Affiliations:** 1Department of Microbiology and Immunology, Faculty of Tropical Medicine, Mahidol University, Bangkok 10400, Thailand; 2Department of Clinical Tropical Medicine, Faculty of Tropical Medicine, Mahidol University, Bangkok 10400, Thailand; 3Mahidol Oxford Tropical Medicine Research Unit, Faculty of Tropical Medicine, Mahidol University, Bangkok 10400, Thailand; 4Hospital for Tropical Diseases, Faculty of Tropical Medicine, Mahidol University, Bangkok 10400, Thailand; 5Department of Immunology, Faculty of Medicine Siriraj Hospital, Mahidol University, Bangkok 10700, Thailand

**Keywords:** chikungunya, virological, serological, clinical analysis, phylogenetic analysis

## Abstract

From 2018 to 2020, the Chikungunya virus (CHIKV) outbreak re-emerged in Thailand with a record of more than 10,000 cases up until the end of 2020. Here, we studied acute CHIKV-infected patients who had presented to the Bangkok Hospital for Tropical Diseases from 2019 to 2020 by assessing the relationship between viral load, clinical features, and serological profile. The results from our study showed that viral load was significantly high in patients with fever, headache, and arthritis. We also determined the neutralizing antibody titer in response to the viral load in patients, and our data support the evidence that an effective neutralizing antibody response against the virus is important for control of the viral load. Moreover, the phylogenetic analysis revealed that the CHIKV strains we studied belonged to the East, Central, and Southern African (ECSA) genotype, of the Indian ocean lineage (IOL), and possessed E1-K211E and E1-I317V mutations. Thus, this study provides insight for a better understanding of CHIKV pathogenesis in acute infection, along with the genomic diversity of the current CHIKV strains circulating in Thailand.

## 1. Introduction

Chikungunya is a re-emerging neglected Tropical disease that has caused several outbreaks throughout history in numerous countries. Chikungunya virus (CHIKV) is transmitted through the bite of an infected *Aedes* mosquito—either *A. albopictus* or *A. aegypti* [1]. It is known to cause persistent arthralgia, which has a significant impact on the patient’s quality of life and has contributed to the economic burden of many countries over the past decades [2,3]. CHIKV is believed to have been originated in Africa in 1952 and is wide-spread in many countries, including Asia [4]. CHIKV exists in three different genotypes—West African (WA), East/Central/South African (ECSA), and Asian [1]. The first reported case of CHIKV infection in Thailand was caused by an Asian genotype in 1953 and spread to many regions of the country, especially Southern Thailand, which had a large outbreak in 2008–2009. The outbreak was caused by the ECSA genotype and infected over 50,000 patients. In 2018–2020, the infection re-emerged with a mutated form of the ECSA genotype, known as India Ocean lineage (IOL), which is responsible for most cases in Southern Thailand, notably in tourist attraction sites [5,6,7]. These outbreaks became a major contributor to the travel-associated imported CHIKV cases, especially from Thailand to other countries [7]. Clinical manifestations of CHIKV infection are high-grade fever, severe arthralgias, myalgias, maculopapular rash, nausea, vomiting, and headache during the acute stage. In the chronic stage, relapsed symptoms with inflammatory polyarthritis and neurological complications have been observed in many patients [8]. However, CHIKV-induced chronic arthritis remains poorly understood and treatment relies solely on the assumption of immunopathogenesis following CHIKV infection [9]. The disease is often underdiagnosed because of the similar clinical presentations to dengue and other endemic infections [10]. Therefore, there is an urgent need for an early and accurate diagnosis of CHIKV for timely treatment and to limit further transmission of the disease. Currently, there are no specific therapeutics against CHIKV infection, and severe persistent symptoms in the chronic stage are found to be associated with a high viral load during the acute period [11,12]. The relationship between viral load and clinical symptoms of CHIKV infection has been documented in a few studies [13,14,15], but the understanding is still limited especially in an endemic area such as Thailand. Hence, we used real-time quantitative RT-PCR to detect and quantify CHIKV viral load in the serum samples of suspected CHIKV-infected patients presented at the Hospital for Tropical Diseases, Mahidol University. The correlation of CHIKV viral load with the clinical symptoms and antibody profiles of the patients was further analyzed. Neutralizing antibodies against CHIKV have been proposed as one of the factors that are associated with a decrease in viral load, providing protection against chronic arthritis in the future [16,17]. Thus, this study also investigates the neutralizing antibody pattern in CHIKV-infected patients and further correlates it with viral load and clinical symptoms. The results from this study will provide insight into clinical manifestation in the correlation between the viral load and antibody profile during CHIKV acute infection.

## 2. Materials and Methods

### 2.1. Ethics Statement Approval

This study was approved by the ethical committee of the Faculty of Tropical Medicine, Mahidol University, Thailand (approved number: FTM ECF-035-01).

### 2.2. Study Population

A total of 214 serum samples of CHIKV-suspected patients from the Hospital of Tropical Diseases, Mahidol University, were included in this study. Suspected cases of CHIKV were defined as the presence of fever, headache, rash, myalgia, and arthralgia by the clinician. Laboratory confirmation was performed by real-time RT-PCR and lateral flow immunoassay (SD biosensor, Gyeonggi-do, Korea) for specific IgM and IgG detection. The sensitivity of the lateral flow immunoassay is 100% for both IgM and IgG with a high specificity of 97.7% for IgM and 99.6% for IgG. In this study, 162 out of 214 cases were confirmed as acute CHIKV-infected cases based on the presence of CHIKV viral RNA and/or CHIKV-specific IgM in the serum samples.

### 2.3. Determination of CHIKV Viral Load

For the quantitative real-time RT-PCR, the RNA from an isolated CHIKV virus strain TM009 2019 was extracted using an RNA extraction kit (Bioneer Accuprep viral RNA extraction kit, Daejeon, Korea) according to the manufacturer’s protocol. The Nsp1 fragment of CHIKV RNA was amplified using CHIKV forward and reverse primers (Appendix A). Nsp1 CHIKV cDNA products were then introduced into the pGem-4z plasmid vector (Promega, Madison, USA) containing a T7 promoter by restriction enzymes—HindIII and Xbal (Promega, Madison, WI, USA)—and transformed into competent JM109 *Escherichia coli*. The recombinant plasmids were then extracted using the Geneaid™ Midi Plasmid Kit (PI025) (New Taipei City, Taiwan) and were purified with a MEGAquick-spin Plus Fragment DNA Purification Kit (Intron Biotechnology, Gyeonggi-do, Korea). The in vitro transcription of the plasmid DNA was performed using the MEGAscript T57 kit (Invitrogen, Carlsbad, CA, USA), according to the manufacturer’s instruction. The synthesized RNA was then purified with the phenol−chloroform extraction method. Purified RNA was quantified by nanodrop (Thermo scientific, Waltham, MA, USA) and the concentration, in nanograms, was converted into RNA copy numbers [18] and used as CHIKV RNA standard, ranging from 10^1^ to 10^10^ copies/µL. Real-time RT-PCR was performed using serially diluted RNA standards. The cycle threshold (Ct) values obtained were plotted against the log dilution of in vitro transcripted RNA to generate the standard curve.

Viral RNA extraction from the serum samples was performed using an RNA extraction kit (Bioneer AccuPrep^®^ viral RNA extraction kit, Daejeon, Korea) according to the manufacturer’s instruction. Viral load estimation was performed by the TaqMan quantitative real-time RT-PCR method. Specific primers and the probe targeting 1200 bp region of the Nsp1 gene of CHIKV were used for amplification (Appendix A). A 20 μL volume reaction was subjected to thermal conditions as follows; activation of reverse transcription at 50 °C for 15 min, followed by initial denaturation at 95 °C for 30 s, 35 cycles of denaturation at 95 °C for 10 s, and annealing at 60 °C for 30 s. All of the experiments were performed on the CFX96 touch real-time PCR machine (Bio-Rad, Hercules, CA, USA). Cycle threshold (Ct) values were plotted against log dilutions of the RNA generated by in vitro transcription for construction of the standard curve. The copy numbers in the serum samples were calculated from the intersection points on the standard cure. Patients’ clinical symptoms were collected using case record forms (CRFs).

### 2.4. Virus Isolation and Propagation

A total of 13 patient serum samples with a high CHIKV viral load (>100,000 copies/mL) were subjected to virus isolation. Briefly, 200 μL of serum samples were inoculated into each well of a 24-well plate containing a confluent Vero cell (kindly provided by Dr. Stephen Whitehead, NIAID, NIH, MD, USA) monolayer. The plates were incubated at 37 °C in a CO_2_ incubator for 90 min, followed by the addition of 500 µL growth medium per well. The plates were then incubated at 37 °C in a CO_2_ incubator for 5 days. After CPE (cell cytopathic effect) was observed in the monolayer, the CHIKV-containing supernatant was collected, aliquoted, and stored at −80 °C for further experiments.

### 2.5. Neutralizing Assay

A plaque reduction neutralization test (PRNT) was performed to determine a CHIKV neutralizing antibody titer of laboratory-confirmed CHIKV cases. The patients’ serum samples were heat-inactivated at 56 °C and diluted to a 1:10 ratio with serum diluent OptiMEM (Invitrogen, New York, NY, USA) supplemented with 0.3% human serum albumin (Sigma-Aldrich, St. Louis, MO, USA). Inactivated serum samples were further diluted into four-fold dilution in 96-well plates. The CHIKV virus (final concentration of 1000 pfu/mL) strain of the ECSA genotype was added to the serum and was incubated at 37 °C for 30 min in a CO_2_ incubator. The cell culture medium was removed from a 90% confluent Vero cell monolayer on 24-well plates and the serum−virus mixture was added to duplicate wells. Cell monolayers were incubated at 37 °C for 60 min in a CO_2_ incubator and an overlay with 0.5% methylcellulose (Sigma, St. Louis, MO, USA) in OptiMEM-GlutaMAX (Thermo Scientific, Waltham, MA, USA) supplemented with 2% FBS (Gibco™ fetal bovine sera, Thermo Scientific, Waltham, MA, USA). The samples were incubated at 37 °C for 4–5 days and the plaques were visualized by staining with 0.2% crystal violet (Sigma-Aldrich, St. Louis, MO, USA). PRNT50 titers were calculated based on a 50% reduction in plaque counts (PRNT_50_) using the curve-fitting method [19] developed by NIH/NIAID (https://bioinformatics.niaid.nih.gov/plaquereduction), accessed date: 20 March 2022.

### 2.6. Sequencing and Phylogenetic Analysis

Viral RNA was extracted from CHIKV virus culture supernatants using a viral RNA extraction kit (Bioneer AccuPrep^®^ viral RNA extraction kit, Daejeon, Korea) according to the manufacturer’s instructions. One-step RT-PCR was performed to amplify the CHIKV envelope 1 (E1) genes with two sets of CHIKV E1 primers (Appendix A) [20,21]. Next generation sequencing was employed to obtain the CHIKV E1 gene segment. The nucleotide sequences from each of the CHIKV isolates were assembled into contigs using Bioedit software (version 7.2.5; http://bioedit.softward.informer.com, assessed on 13 February 2022). The nucleotide sequences described in this study were deposited into the GenBank database (Appendix A). The obtained sequences were aligned with reference CHIKV sequences retrieved from the GenBank database. The phylogenetic trees were constructed based on the Maximum-likelihood method using the Tamura-3 parameter (T92) in the Mega 6 software [22]. The sequences were also translated into protein sequences and aligned with CHIKV reference protein sequences from GenBank to observe the presence of E1 mutations (V226A, I317V, and K211E). The E1 amino acid positions were annotated following the CHIKV E1 protein (Gene Bank Accession no. ADZ47902-India-2006) [23].

### 2.7. Statistical Analysis

The IBM SPSS statistic program (version 23.0, IBM) was used for statistical analysis in this study. Descriptive statistics were used to demonstrate the demographic data of the patients, a Student’s *t*-test was used to investigate the association between viral load and clinical features of the patients, and *p* values less than 0.05 were considered to be statistically significant. Pearson Chi-square test was used to test for correlation in the clinical features vs. categorical variables such as age group. A Mann−Whitney test was used to compare PRNT_50_ titers among the clinical symptoms, gender, and age groups. *p* values less than 0.05 were considered to be statistically significant.

## 3. Results

### 3.1. Demographic Data

The clinical features of suspected CHIKV-infected patients who presented at the Hospital for Tropical Diseases, Mahidol University, from October 2019 to December 2020 were recorded. Among these CHIKV suspected patients, 162 out of 214 patients were confirmed by laboratory procedure to have acute CHIKV infection, either through the presence of CHIKV viral RNA or CHIKV-specific IgM, or the presence of both CHIKV viral RNA and CHIKV-specific IgM in the serum samples. Quantitative real-time RT-PCR targeting a CHIKV Nsp1 gene was performed [24]. The limit of detection in this study was 100 copies/mL; 77 out of 214 serum samples had CHIKV viral RNA greater than 100 copies/mL. On the other hand, 113 out of 214 serum samples were CHIKV IgM positive by lateral flow immunoassay. Among the acute laboratory-confirmed CHIKV patients, 57.4% were female and 42.6% were male. The majority of patients were of Thai nationality (95.1%), between the age of 31 to 45 (30.9%) and 45 to 60 years old (33.3%), and 91 out of 162 (56.2%) reported no underlying diseases (Table 1).

### 3.2. CHIKV Viral Load and CHIKV Antibody Profile in the Patient Serum Samples

Among these CHIKV laboratory-confirmed cases, 162 cases were considered acute febrile cases (fever < 10 days), and 11 cases had a persisting fever for 10–21 days. The number of RT-PCR positive samples were high during day 1 to 7 of illness onset and gradually declined after day 5, with no CHIKV viral RNA observed in the serum samples after day 14 (Figure 1). In some cases, CHIKV-specific IgM and IgG were detected on day 1 and IgM remained detectable in six patients after 14 days of illness onset. However, CHIKV specific IgM and IgG were found in most of the patient’s serum samples collected on day 3 to day 7 of illness onset. Positive RT-PCR results were inversely correlated with IgM and IgG. The presence of CHIKV antibodies was found to cause a decrease in the viral load in many patients, especially on days 5 to 14 of illness onset. Additionally, the presence of CHIKV-specific IgG in some patients showed evidence of past exposure to CHIKV infection (Figure 1). Among the 77 patients with CHIKV viral RNA greater than 100 copies/mL (RT-PCR positive), 28 were positive for CHIKV IgM and 7 were positive for CHIKV IgG. All 85 patients with viral RNA less than 100 copies/mL were positive for CHIKV IgM and 26 were positive for CHIKV IgG (Table 2).

### 3.3. Clinical Features, Hematological Profile, and Viral Load Analysis

The clinical features observed among CHIKV laboratory-confirmed cases were varied due to the variation in the sample collection time points. The most common features among the patients were fever (93.8%), arthralgia (81.5%), maculopapular rash (51.2%), headache (24.1%), and arthritis (17.3%). Only 0.6% of cases reported a neurological complication (Table 3). The association between the clinical features and mean CHIKV viral load was analyzed. The results appeared to have a significant difference in mean viral load between patients with fever and without fever (*p* < 0.05), headache and without headache (*p* = 0.028), and arthritis and without arthritis (*p* = 0.04). The most frequently reported symptoms during days 1–7 of illness onset were fever (41%, 67 out of 162) and arthralgia (49%, 79 out of 162). Among them, 69.7% (113 out of 162) of patients reported the combination of fever and arthralgia during days 1–7 of illness onset. The manifestation of other symptoms such as headache and arthritis also appeared between days 1 and 7, while maculopapular rash was predominant between days 5 to 7 of illness onset (35%, 56 out of 162) (Appendix A).

In this study, we also analyzed the association between age and clinical features. No significant association was observed between clinical features among different age groups, except maculopapular rash (*p* = 0.014) (Table 4). To further determine the hematological profile of CHIKV patients, we followed the complete blood count reference values for healthy Thai adults established by the Faculty of Medicine Siriraj Hospital, Mahidol University, Bangkok, Thailand [25]. We compared the hematological profile among confirmed acute CHIKV-infected cases (N = 162) of different genders and the median value was used to measure the parameters (Table 5). Although the median values were within the normal reference range for both groups, the leukocytes level was found to be at the lower end of the reference range in both genders, while the neutrophil level of the patients was found to be at the higher end of the reference range. The difference in hemoglobin, hematocrit, and platelets between males and females appeared to be statistically significant (*p* < 0.05) (Table 5).

### 3.4. Antibody Responses against CHIKV Virus in Infected Patients

To further investigate the pattern of viral load and antibody profile in confirmed CHIKV patients (*n* = 162), apart from the detection of IgM and IgG by the lateral flow immunoassay, we determined the CHIKV neutralizing antibody titers in these serum samples, which have an important inhibitory function against CHIKV. We were able to perform a neutralizing antibody assay on 159 out of 162 patients because of the limited availability of patient serum samples. An inverse correlation between viral load and neutralizing antibody titers was observed in this study (Figure 2A). A high viral load (1 × 10^5^ copies/mL) was detected in the serum samples collected during days 1–8 of fever. A significant reduction in CHIKV viral load was observed in the samples collected 8 days after illness onset (Figure 2A). In contrast, increased levels of neutralizing antibody titers were detected in the samples collected after 6 days and the highest levels of neutralizing titer were observed in the samples collected on day 16 of illness onset. High neutralizing antibodies (PRNT_50_ titer > 1:1000) were found in 72% (114/159) of patients. In addition, we further analyzed the CHIKV viral load in two groups of samples—patients with CHIKV neutralizing antibodies (PRNT_50_ titer > 1:10) and without neutralizing antibodies (PRNT_50_ titer < 1:10). The results showed that the viral load was significantly lower in the group with a high neutralizing antibody titer (Figure 2B; *p* < 0.0001). Further analysis of the immune response in CHIKV-infected patients showed that the CHIKV neutralizing antibody titer was higher in patients with prominent symptoms such as fever, arthralgia (*p* ≤ 0.01), arthritis, and maculopapular rash (*p* ≤ 0.01) (Figure 3A). However, headache was found to be associated with a low neutralizing antibody titer (*p* ≤ 0.01). Additionally, there was no significant difference between CHIKV neutralizing antibody titers and gender and age groups (Figure 3B,C).

### 3.5. CHIKV Sequence Analysis

To monitor the genomic characterization of CHIKV, the CHIKV partial E1 gene (about 1.2 kb) was amplified from 13 patients’ sera with a high viral load. Sequencing of the partial E1 gene was performed and the evolutionary characteristic of the CHIKV isolates in this study (Appendix A) were analyzed along with global strains by phylogenetic analysis. The results showed that the CHIKV isolates from the present study belonged to the Indian Ocean lineage within the ECSA genotype and showed a high identity with CHIKV strains circulating in Thailand during the large outbreak in 2018–2020, including strains from the outbreak in India in 2016, Pakistan in 2016, and Bangladesh in 2017. The current viruses were also grouped with strains from travelers returning to Australia from India in 2016, an outbreak in China in 2019, Taiwan in 2019, and imported CHIKV cases in Myanmar in 2019. However, these strains did not cluster with the CHIKV strains isolated from the first large outbreak in Thailand during 2008 to 2009 (Figure 4A). The amino acid substitutions of the current strain were also analyzed, focusing on the E1 protein, and comparing it with the past strains from large outbreaks in Thailand as well as outbreaks in other countries—India, Bangladesh, Italy, and China (Figure 4B). We found that none of the CHIKV strains from the current study contained the E1-A226V mutation that was present in the Thai strains during the 2008–2009 outbreak. Instead, the current strains carried E1-K221E and E1-I317V mutations, which are distinct mutations present in a new sub-lineage that separated from the past IOL CHIKVs and are believed to have been introduced to Bangladesh in around late 2015 and Thailand in early 2017.

## 4. Discussion

Although CHIKV infection is an emerging mosquito-borne virus responsible for outbreak in many countries, the progression of prolonged chronic arthralgia/arthritis in infected patients continues to remain a burden to patients. This prompted many scientists not only to develop possible treatments to combat and prevent the spread of disease, but also to understand the pathogenesis of disease in progression to chronic conditions. Chronic stage CHIKV is defined as having persistent symptoms of arthralgia/arthritis for more than 3 months after the initial diagnosis [26]. According to a few prior studies, CHIKV IgM was detectable in patients after 2–3 years [26,27,28] and the longest persistent case of joint pain and stiffness was found in South African patients after 3 to 5 years of acute infection [26]. So far, the underlying cause of chronic CHIKV arthropathy remains unclear [29], and several studies have reported the factors that could contribute to helping alleviate chronic arthralgia/arthritis. The main contributors to chronic infection are viral load and individual immune responses [16,17,30].

Among the 214 patients who presented at the hospital for Tropical Diseases from October 2019 to December 2020, 162 patients were confirmed to have CHIKV infection. The majority of viremia cases were observed on days 1 to 5 of illness onset, along with detectable IgM antibodies after day 4. This explains the peak of the viral load during symptom onset, which leads to activation of the host immune response [31,32,33,34]. In line with a few of the studies reported in the past, we also detected persistent IgM in some patients on day 21 of illness onset [27,28]. Similar to other findings, our study also detected CHIKV RNA in up to 30% of patients with the presence of IgM and IgG [32,35]. Furthermore, neutralizing antibodies were observed as early as day 1 in one patient with a PRNT_50_ titer of >1:100, which highlights the early presence of neutralizing antibodies in CHIKV-infected patients, which could indicate residual antibodies from prior infection. In concordance with a study done in India [17], the results from our study also showed an inverse correlation between CHIKV neutralizing antibody titer and viral load, which provides evidence of the host immune response defending against the virus.

The most frequently observed clinical symptoms in this study were fever, arthralgia, headache, and the presentation of skin rash, which are the common features of acute CHIKV infection [9,13,32,36,37]. Furthermore, the majority of patients in this study were either from the middle-aged (31–45) or old adult group (46–60), so the symptoms of CHIKV infection appeared more in those age groups compared with children and young adults. In addition, the majority of the patients in this study were 31–45 years old, so CHIKV infection was more prevalent among this group in this study compared with the younger age groups. We also observed that a higher viral load was significantly associated with the development of fever, headache, and arthritis. A similar study was also reported from Thailand, where a significant association between viral load and arthralgia was found [37]. However, because of the difference in sample sizes, our results were not aligned with their findings. Additionally, a study conducted in India indicated that myalgia is associated with a higher viral load in the acute phase of CHIKV infection in children [13]. Currently, there are limited data showing the relationship between viral load and clinical symptoms in endemic areas. Therefore, more studies need to be conducted for a better understanding of these associations. Because our study is a cross-sectional study using the leftover serum samples from the routine hospital diagnosis, there are some missing clinical and demographic data. In addition, the time lapse between symptom onset and the actual date that the patients presented at the clinic might have also affected the acute phase reporting. We also compared the CHIKV neutralizing antibody titer in relation to the clinical symptoms and observed that patients with CHIKV-predominate symptoms, such as arthralgia, arthritis, and skin rash, developed higher neutralizing antibody titers compared with those without the symptoms. This highlights the importance of the immunopathology of CHIKV disease, where the development of clinical symptoms is caused by the activation of the host immune response to the virus, especially in the acute phase of infection, as symptoms are dominated during days 1 to 7 of illness onset. As the neutralizing antibody increased after day 10 of illness onset, fewer symptoms were found in infected patients. Our study had some limitations; the statistical analysis in this study was performed using data at a single time point, which may have affected the interpretation of results. Therefore, a prospective study with well controlled parameters and a larger sample size are needed to confirm the association between viral load, antibody responses, and clinical symptoms.

Apart from understanding viral load and antibody response in patients, studies of the genetic variation of the current CHIKV strain circulating in Thailand are crucial for controlling disease outbreaks. In concordance with previously reported studies, the CHIKV strain isolated from patients in the present study also belongs to the ECSA genotype, which is the same genotype as the outbreak in 2008–2009 in Thailand [38,39,40]. Our results also revealed that none of the CHIKV isolates possessed the E1-A226V mutation, which is known to enhance the fitness, infectivity, dissemination, and transmissibility of CHIKV in *A. albopictus* mosquitoes [41]. Notably, we observed that the present isolated strains harbored E1-K211E and E1-I317V, which is consistent with the recently documented studies in Thailand [37,39,42,43]. Other studies have demonstrated that the E1-K211E/E2-V264A mutation has a higher fitness for *A. aegypti* compared with parental E1-A226V strains adapted in *A. albopictus*. This increases the efficiency of CHIKV circulating in *A. aegypti* endemic areas, which are prevalent around Thailand, whereas an abundance of *A. albopictus* is restricted in the southern part of Thailand [44,45]. Further studies are needed to evaluate the functional role of the E1-I317V mutation in adaptation to the mosquito species. An evolutionary study of the CHIKV IOL lineage suggested that E1-A226V mutation was absent from India after 2014 and a new amino acid substitution of E1-K211E/E2-V264A emerged along with an additional mutation of E1-I317V in early 2010, which rapidly spread to Bangladesh in 2017 and from Bangladesh to Thailand in 2018–2020 [43]. Meanwhile, an analysis of the genetic diversity of field-catch *A. aegypti* mosquitos in endemic areas of Thailand also detected the presence of the E1-K211E mutation [40]. This evidence supports the idea that the efficient transmission of CHIKV in different vectors plays an essential role in the evolution of CHIKV over a period of time. Overall, the results from our study show that CHIKV circulating in Bangkok, Thailand, in 2019–2020 belongs to the ECSA genotypes and carries E1-K211E and E1-I317V mutations.

## Figures and Tables

**Figure 1 viruses-14-01805-f001:**
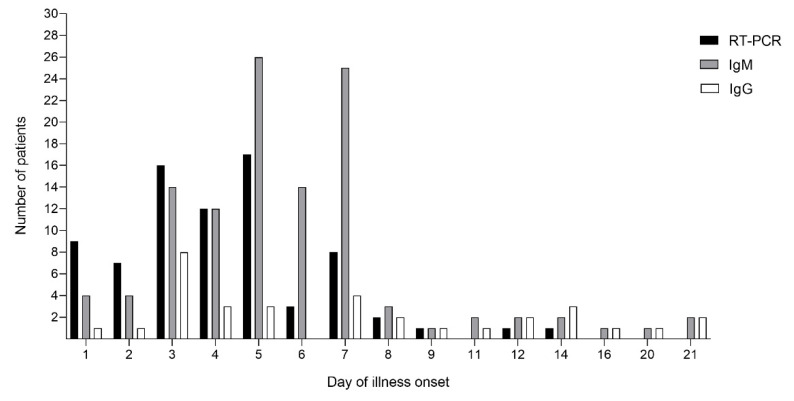
Results of the CHIKV lateral flow immunoassay and RT-PCR confirmation in patients (*n* = 162). Percentage of positive RT-PCR, CHIKV specific IgM, and IgG tests per day of illness onset.

**Figure 2 viruses-14-01805-f002:**
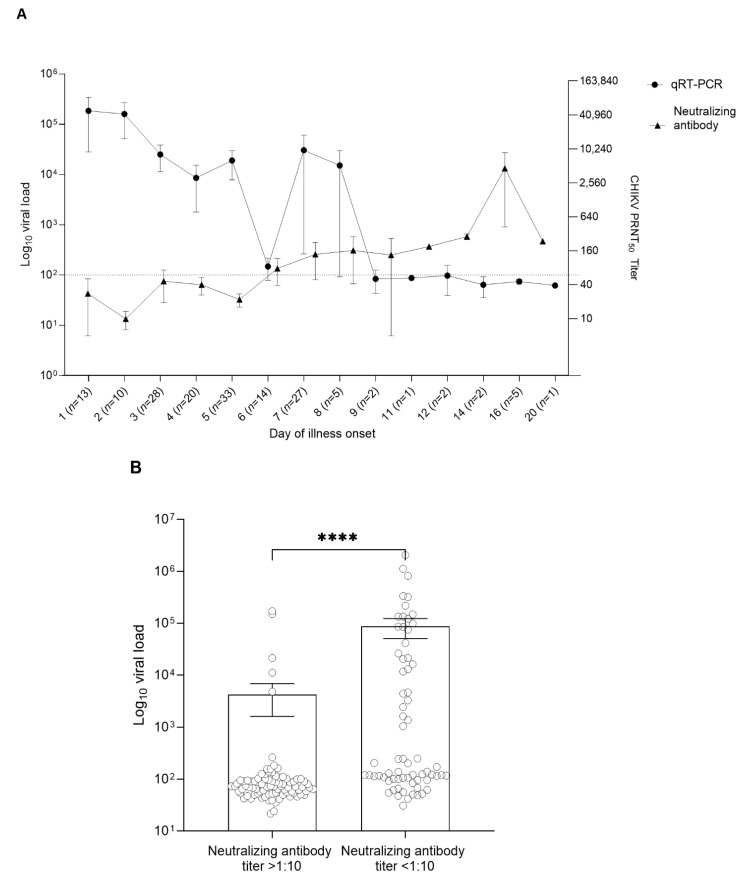
Immune response in patient samples with confirmed CHIKV infection (*n* = 159). (**A**) Dynamic of the CHIKV viral load and neutralizing antibodies on the day of illness onset. *X*-axis shows the day of illness with the number of patients. (**B**) Comparison of CHIKV viral load between the neutralizing antibody titer > 1:10 and neutralizing titer < 1:10. Error bars showing the standard mean error of the sample means. *p*-values are calculated using Mann−Whitney test. **** *p* < 0.0001.

**Figure 3 viruses-14-01805-f003:**
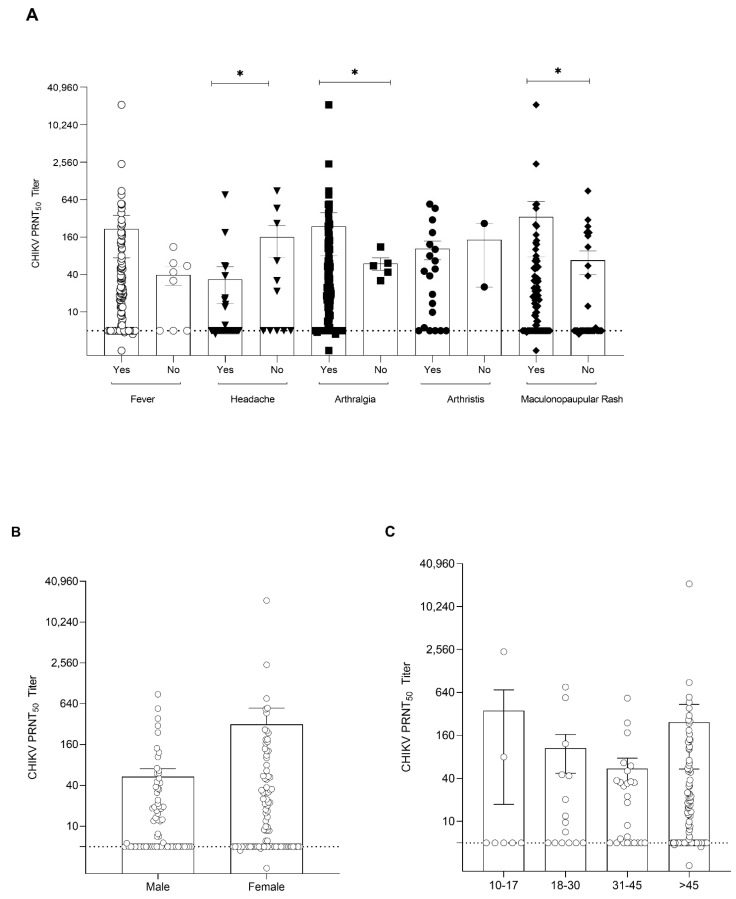
Association of CHIKV neutralizing antibodies vs. clinical features (**A**), gender (**B**), and age groups (**C**). Error bars showing the standard deviation of the sample means. *p*-values were calculated using the Mann−Whitney test. * *p* < 0.01.

**Figure 4 viruses-14-01805-f004:**
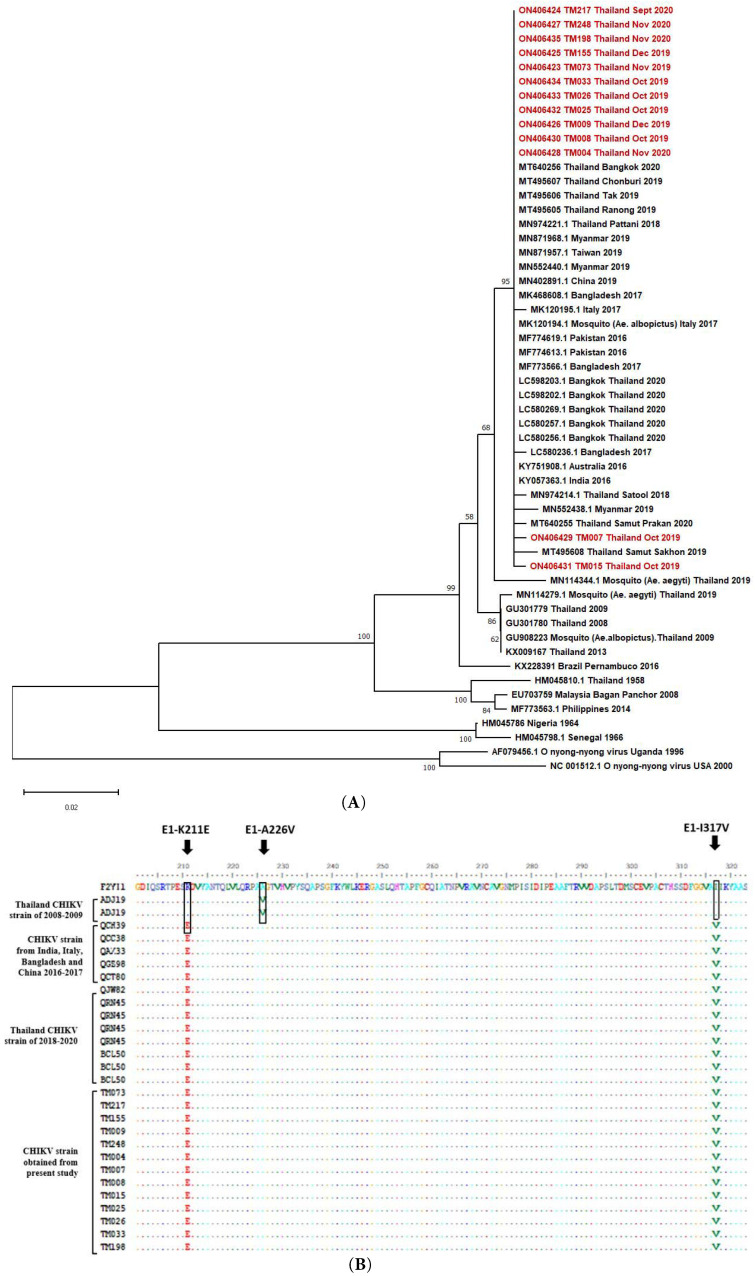
Sequence analysis of the isolated CHIKV strains of patients. (**A**) Phylogenetic analysis of partial CHIKV E1 protein using the maximum-likelihood method with 1000 bootstrap replicates represented at the brand notes. CHIKV strains isolated from this study are shown in red text. (**B**) Alignment of amino acid sequences of isolated CHIKV strains in this study.

**Table 1 viruses-14-01805-t001:** Demographic data of the CHIKV-infected patients (N = 162).

	Patient Characteristics	Patients, N (%)
Gender	Male	69 (42.6)
	Female	93 (57.4)
Age	10 to 17	8 (4.9)
	18–30	31 (19.1)
	31–45	50 (30.9)
	46–60	54 (33.3)
	60+	19 (11.7)
Race	Thai	154 (95.1)
	Myanmar	4 (2.5)
	Laos	1 (0.6)
	Cambodia	1 (0.6)
	Others	2 (1.2)
underlying diseases	Yes	61 (37.7)
	No known underlying disease	91 (56.2)
	No data	10 (6.2)

**Table 2 viruses-14-01805-t002:** Results of the CHIKV lateral flow immunoassay and RT-PCR confirmation in patients (*n* = 162).

Parameters		RT-PCR		Total
		>100 Copies/ml	<100 Copies/ml	
CHIKV IgM	Positive	28	85	113
	Negative	49	0	49
	Total	77	85	162
	Positive	7	26	33
CHIKV IgG	Negative	70	59	129
	Total	77	85	162

Positive for RT-PCR is defined as more than 100 copies/mL.

**Table 3 viruses-14-01805-t003:** Association of viral load and clinical features among CHIKV-infected patients (N = 162).

Clinical Features	Presence of Symptom	Log_10_ Viral Load	*p*-Value
	(%)	Mean ± SD	
Fever	Yes (93.8)	5.76 ± 2.51	**<0.05**
	No (4.9)	4.38 ± 0.38	
Headache	Yes (24.1)	6.83 ± 3.46	**0.028**
	No (8.0)	5.06 ± 1.94	
Arthralgia	Yes (81.5)	5.90 ± 2.85	0.805
	No (2.5)	6.26 ± 3.52	
Arthritis	Yes (17.3)	6.32 ± 3.42	**0.04**
	No (3.7)	4.90 ± 0.33	
Maculopapular rash	Yes (51.2)	5.33 ± 2.30	0.271
	No (20.9)	5.96 ± 2.96	
Neurological complications	Yes (0.6)	3.75 ± 0.0	0.393
	No (3.1)	7.91 ± 3.97	

Note: Student’s *t*-test was used to calculate *p*-values. Viral load is presented in the log^10^ value. Bold numbers indicate statistically significant *p*-values.

**Table 4 viruses-14-01805-t004:** Clinical features observed in different age groups of CHIKV-infected patients (N = 162).

Clinical Manifestations	All Patients *n* = 162	Age Group					*p*-Value
		Children ^1^ *n* = 8	Young Adults ^2^ *n* = 31	Middle-Aged Adults ^3^ *n* = 50	Old Adults ^4^ *n* = 54	Seniors ^5^ *n* = 19	
Fever	152 (93.8)	8	29	48	48	19	0.324
Headache	39 (24.1)	3	7	13	12	4	0.464
Arthralgia	132 (81.5)	6	21	41	47	17	0.305
Arthritis	28 (17.3)	3	7	6	9	3	0.664
Maculopapular rash	83 (51.2)	5	20	31	23	4	0.014
Neurological complications	1 (0.6)	0	0	1	0	0	0.301

^1^ Age 10–17 years old, ^2^ Age 18–30 years old, ^3^ Age 31–45 years old, ^4^ Age 46–60 years old, ^5^ Age 60+ years old. Bold numbers indicate statistically significant *p*-values.

**Table 5 viruses-14-01805-t005:** Hematological findings in CHIKV-infected patients (N = 162).

Parameters	Male *n* = 69	Female *n* = 93	*p*-Value
Leukocytes/${\mathrm{cumm}}^{3}$	4.5 (1.50–12.6)	5.0 (2.4–14.6)	0.893
Hemoglobin (g/dL)	14.4 (11.0–17.0)	13.3 (8.8–15.5)	<0.05
Lymphocytes (%)	22.5 (3.0–53.0)	24.7 (4.0–56.0)	0.290
Neutrophils (%)	61.9 (3.6–85.0	62.0 (32.0–90.0)	0.896
Eosinophils (%)	1.0 (0–18.0)	1.0 (1.0–11.0)	0.689
Monocytes (%)	5.0 (1.0–22.0)	4.2 (1.0–10.0)	0.084
Basophils (%)	0 (0–7.0)	0 (0–2.0)	0.291
Hematocrit (%)	43.1 (32.9–52.7)	39.7 (14.0–46.6)	<0.05
Platelets/${\mathrm{cumm}}^{3}$	174.5 (4.0–507.0)	216.5 (33.0–495.0)	0.009

Hematological profiles are presented as medians and ranges. CHIKV positive is defined as positive for either RT-PCR test (>100 copies/mL), IgM lateral flow immunoassay, or positive for both tests. Statistical *p*-values between the groups were obtained from ANOVA.

## Data Availability

Data is contained within article and Supplementary Material.

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
