# Peer review of "Virological, Serological and Clinical Analysis of Chikungunya Virus Infection in Thai Patients"

_viruses, 2022, doi:10.3390/v14081805_

Round 1

Reviewer 1 Report

Overall comment:

An interesting manuscript, which will require proof reading for spelling & grammatical errors before publication. There are statements in the Methods, Results & Discussion that will require changes/explanations/revisions as below

Abstract: acceptable

Introduction: acceptable

Methods:

Line 74: “….214 serum samples ……” does this equate to the 214 unique patients in the Results ?

Line 130 – was the ESCA strain used in the neutralizing assay closely similar to the strain detected in the CHIK RNA positive samples ?

Line 156 – why did you use a strain from 2006 to annotate your sequences?

Results:

Line 172 …. “162 of 214 patients had acute CHIKV infection by PCR or IgM serology”, therefore 52 cases were clinically diagnosed ? Could these not be dengue, zika or some other etiology ? If so are these cases included in your later analysis as it’s not clear.

Line 178 … “among 162 samples, 77 were PCR positive and 113 were IgM pos and 28 were both PCR & IgM pos which totals 218, which is more than the 162 at the outset – suggest that you tabulate for clarity, instead of Fig 1B.

Line 179  - for those 85 IgM pos and CHIK PCR negative samples how did you exclude or account for samples that could be a false-pos for CHIK IgM antibody ?

Fig 1A – difficult to tell RT-PCR column from IgM column as both have similar shades

Table 2 – When I “eyeball” the mean differences in the viral loads for the presence or absence of symptoms for fever & headache they are not substantially different eg 5.76 vs 4.38 is about 1 log. Interestingly for arthralgia the mean VL is higher for absence of symptoms (6.26) vs presence (5.90), how do you explain this reversal ?

Given that samples would have been collected at various times after disease onset, which means that the viral load could vary coupled with the patients having a diverse age range, all of which could have an impact on viral load, can one convincingly suggest there is an association ?

Table 4 – the mean ranges of haemoglobulin for both males and females in the infected group is within the accepted ranges for normal individuals so I am unclear why there is a statistical association?

Figures 2A, 3A & B are extremely small and difficult to read and will have to be much larger. For Figure 2A on the x axis, what do the bracketed numbers refer to ?

Line 259 – this statement “Neutralizing antibodies ……” is unclear and requires revision

Figure 4 – again both are incredibly tiny and not readable to someone over 50 yrs

Discussion:

Line 329 you state that arthritis is a global threat”. Why is it a global threat as its often reversible and the condition is not transmissible, more of a temporary disability ?

Line 333 – you state CHIK is a chronic infection yet the longest period of detectable viraemia was 21 days. Contrast that with hepatitis B or C where patients remain infected/infectious for decades and are therefore legitimately chronically infected ?

Line 340 – IgM antibody can be detected up to 5 or 6 mos after many infections and often depends upon the sensitivity of the testing platform/assay. In some infections, such as West Nile virus, it is well documented that IgM can be detectable for up to 2 or more years, so 21 days is not that excessive, is it  ?

Line 343 – you state that some patient’s had neutralizing antibody at day 1 or thereabouts, could these patients not have residual antibody from a prior infection ?

References: sufficient

Author Response

Dear Reviewer,

Thank you for giving us the opportunity to submit a revised draft of our manuscript titled "Virological, serological, and clinical analysis of chikungunya virus infection in Thai patients". We truly appreciate the time and effort you and the reviewers dedicated to providing valuable feedback on our manuscript. We appreciate the reviewers’ insightful response, and we have been able to incorporate the changes as per the reviewers’ suggestions.

Enclosed please find a point by point response to the reviewer’s comments and we highlighted the changes in the manuscript.

Thank you for your consideration.

Sincerely,

Kobporn Boonnak, Ph.D.

Department of Immunology

Faculty of Medicine Siriraj Hospital

Mahidol University, Thailand

Phone: 662-306-9173

Fax: 662-643-5583

Reviewer 2 Report

General Comments

This article “Virological, serological and clinical analysis of chikungunya virus infection in Thai patients” provides important information on the circulating strains of CHIKV in Thailand during the outbreak between 2019 and 2020 while determining the relationship between viral load, clinical presentation and serological profile. Below are a few comments/questions.

Method/Results

Demographic data

·         What was the criteria/definition for laboratory-confirmed CHIKV patients; qRT-PCR positive, IgM positive or both?

·         What is the specificity of the antibody detection assay/kit and what is the likelihood of false positives?

·         The statement on underlying diseases should read “no known underlying diseases”

Clinical features

·         Does table 3 show the log of viral load? If yes, it should be stated

·         Based on the inverse correlation between IgM levels and viral load, are the authors saying that those presenting symptoms like headaches where in the early days of infection and those without in the late days like 7 days and after?

·         Did authors collect information on whether or not participants had self-medicated before visiting hospitals?

Discussion

·         Why is CHIKV infection most prevalent in people between the ages of 31-45? is it life style?

·         What is the clinical diagnosis of CHIKV in Thailand and how did the results of the clinical diagnosis compare with that or the authors?

Author Response

(The authors gave the same response as above.)

Round 2

Reviewer 1 Report

My questions have been addressed adequately